# Impact of Cellulose-Rich Organic Soil Amendments on Growth Dynamics and Pathogenicity of *Rhizoctonia solani*

**DOI:** 10.3390/microorganisms9061285

**Published:** 2021-06-12

**Authors:** Anna Clocchiatti, Silja Emilia Hannula, Muhammad Syamsu Rizaludin, Maria P. J. Hundscheid, Paulien J. A. klein Gunnewiek, Mirjam T. Schilder, Joeke Postma, Wietse de Boer

**Affiliations:** 1Department of Microbial Ecology, Netherlands Institute of Ecology (NIOO-KNAW), 6708 PB Wageningen, The Netherlands; m.rizaludin@nioo.knaw.nl (M.S.R.); m.hundscheid@nioo.knaw.nl (M.P.J.H.); p.kleingunnewiek@nioo.knaw.nl (P.J.A.k.G.); 2Department of Terrestrial Ecology, Netherlands Institute of Ecology (NIOO-KNAW), 6708 PB Wageningen, The Netherlands; e.hannula@nioo.knaw.nl; 3Biointeractions and Plant Health, Wageningen University & Research, 6708 PB Wageningen, The Netherlands; mirjam.schilder@wur.nl (M.T.S.); joeke.postma@wur.nl (J.P.); 4Soil Biology Group, Wageningen University, 6708 PB Wageningen, The Netherlands

**Keywords:** bacterial communities, cellulose, damping-off, fungal communities, organic amendments, *Rhizoctonia solani*, saprotrophic fungi, sustainable agriculture, wood sawdust

## Abstract

Cellulose-rich amendments stimulate saprotrophic fungi in arable soils. This may increase competitive and antagonistic interactions with root-infecting pathogenic fungi, resulting in lower disease incidence. However, cellulose-rich amendments may also stimulate pathogenic fungi with saprotrophic abilities, thereby increasing plant disease severity. The current study explores these scenarios, with a focus on the pathogenic fungus *Rhizoctonia solani*. Saprotrophic growth of *R. solani* on cellulose-rich materials was tested in vitro. This confirmed paper pulp as a highly suitable substrate for *R. solani*, whereas its performance on wood sawdusts varied with tree species. In two pot experiments, the effects of amendment of *R. solani*-infected soil with cellulose-rich materials on performance of beetroot seedlings were tested. All deciduous sawdusts and paper pulp stimulated soil fungal biomass, but only oak, elder and beech sawdusts reduced damping-off of beetroot. Oak sawdust amendment gave a consistent stimulation of saprotrophic Sordariomycetes fungi and of seedling performance, independently of the time between amendment and sowing. In contrast, paper pulp caused a short-term increase in *R. solani* abundance, coinciding with increased disease severity for beet seedlings sown immediately after amendment. However, damping-off of beetroot was reduced if plants were sown two or four weeks after paper pulp amendment. Cellulolytic bacteria, including *Cytophagaceae*, responded to paper pulp during the first two weeks and may have counteracted further spread of *R. solani*. The results showed that fungus-stimulating, cellulose-rich amendments have potential to be used for suppression of *R. solani*. However, such amendments require a careful consideration of material choice and application strategy.

## 1. Introduction

Soil-borne plant pathogenic fungi represent a major threat for agricultural production [1]. In modern agriculture with large-scale production of monoculture crops, the negative impact of soil-borne diseases is largely controlled with the use of chemical fungicides. However, fungicides are increasingly eliminated because of concerns regarding human health and negative effects on the environment. In addition, many fungicides are not effective for managing pathogen populations in the soil, and can even select resistant pathogen genotypes [2]. As an alternative to chemical control, attention is given to the development of bio-based, sustainable methods for pest and disease management. One of the options that is widely examined is the use of organic amendments to promote disease-suppressing activities in native soil microbial communities [3,4,5,6].

Incorporated organic materials serve soil microbes as carbon and energy sources, and their addition is often followed by an increased total microbial activity and biomass [7,8]. That often coincides with lower incidence of soil-borne diseases [7,9]. This is ascribed to intensification of competition among microbes for resources, including root exudates and plant debris [10,11]. However, antagonistic activities, such as production of fungicidal compounds and lytic enzymes attacking the fungal cell wall, also can increase, and are particularly important for suppressing germination and growth of pathogenic fungi [5,12]. 

Stimulation of saprotrophic fungi in arable soils could be a promising approach to suppress pathogenic fungi, since they can directly compete for resources or antagonize pathogenic fungi [13], but can also affect the suppression indirectly via stimulation of fungus-suppressing and fungus-feeding bacteria [14,15,16,17]. Saprotrophic fungal biomass is low in intensively managed arable soils, and one of the major causal factors is the limited input of decomposable organic matter [18,19,20]. In an earlier study, it was shown that cellulose-rich materials give a rapid and lasting stimulation of saprotrophic fungal biomass [20]. Therefore, amendments with cellulose-rich materials could be used to increase the involvement of saprotrophic fungi in suppression of soil-borne fungal pathogens.

However, the use of organic amendments for the purpose of disease suppression should be carefully evaluated, since it is known that the effect of organic amendments on soil-borne plant pathogens can strongly vary [21,22]. Such differences in suppression of soil diseases depend on edaphic factors, as well as on the biochemical composition and decomposability of the added materials [21,23]. For instance, it has been shown that mature composts can have disease-suppressive properties, while also carrying a low risk of increasing disease conductivity of soils [21,22,24].

In contrast to mature composts, fresh organic amendments can increase soil conductivity of diseases [10,21]. In fact, many soil-borne pathogens can grow as saprotrophic, and profit from an increased availability of accessible and degradable substrates [25]. Interestingly, an increase in disease conductivity after the addition of fresh organic materials can be followed by an increase in disease control [10,23,26]. Such reduction of disease incidence after an initial increase may be due to a pathogen-induced stimulation of antagonistic microorganisms [27]. Therefore, timing of soil amendment and sowing of crops need to be considered when utilizing fresh organic materials as disease-suppressive amendments.

*Rhizoctonia solani* is a soil-borne pathogenic fungus that causes worldwide diseases in several important crops [28]. There are indications that saprotrophic growth of the pathogen with certain organic amendments can increase the severity of damping-off diseases caused by *R. solani* [23,29]. However, the population dynamics and pathogenicity of *R. solani*, as affected by organic substrates, has received limited attention [29]. *R. solani* is a basidiomycetal fungus with the ability to produce a broad array of plant cell-wall-degrading enzymes [30,31]. These are used for decomposition of cellulose, as part of both necrotrophic and saprotrophic growth of *R. solani* [25,32]. This is considered one of the main reasons why suppression of *R. solani* by organic plant-based amendments is relatively rare, as compared to other fungal pathogens [22,33,34]. Moreover, since *R. solani* can consume cellulose-rich substrates with a broad variety of C:N ratios [35], the control or population dynamics of *R. solani* is not easily predicted based on the quality of organic amendments. Hence, in order to maximize the benefits of cellulose-rich organic amendments in *R. solani* control, it is necessary to understand their impact on both the pathogen and non-pathogenic soil microbes, in relation to their effect on plant performance.

The objectives of this study were to investigate: (1) the growth performance of *R. solani* on different types of sawdusts and paper pulp; (2) the relationship between the growth of *R. solani* on these cellulose-rich materials and their effect on damping-off beetroot seedling disease; (3) the relationship between stimulation of the total soil fungal biomass by cellulose-rich materials and *R. solani* damping-off of beetroot; and (4) the effect of timing of sowing of seeds after the addition of organic materials (cellulose-rich and cellulose-poor) on *R. solani* damping-off of beetroot and other members of the fungal and bacterial community.

We hypothesized that the ability of *R. solani* to grow in cellulose-rich organic materials is inversely related to the effectiveness of the organic materials to suppress *R. solani* damping-off disease. In addition, we hypothesized that a stimulation of the total soil fungal biomass, but not of *R. solani*, by cellulose-rich amendments will coincide with disease suppression. We further hypothesized that for organic materials increasing the *R. solani* population in soil, a time interval between the addition of the organic material and sowing would give a better control of the disease, as a result of stimulation of antagonistic interactions between the pathogenic fungus and other soil microbes.

## 2. Materials and Methods

The growth of *R. solani* on 14 woody substrates and paper pulp was assessed. The experiment consisted of an assay in Petri dishes (named hereafter: *R. solani* performance (RsP)).

### 2.1. RsP Experiment: Preparation of Wood Sawdust and Paper Pulp

The organic materials included in the current study were sawdust obtained from beech, oak, hazel, alder, birch, walnut, maple, elder, holly, willow, hawthorn, snowy mespilus, cypress, and Douglas fir, as well as paper pulp (SCA Hygiene Products Suameer, Sumar, The Netherlands) (Table 1). Beech sawdust was obtained from a local producer (Bemap Houtmeel, Bemmel, The Netherlands). Wood branches of all the other tree species were obtained from trees in the forests near Wageningen in June 2018. Each branch was cut with a chainsaw, and sawdust was collected and further crushed with a cutting mill (SM 100, Retsch, Haan, Germany). All types of sawdust were sieved (2 mm) and stored at room temperature in a dry place until use. Paper pulp, pig hair meal, and shrimp meal were dried and further sieved (2 mm) before use.

### 2.2. RsP Experiment: Assay of Pathogen Performance on Woody Substrates

A portion of 100 g of sawdust of 14 tree species (Table 1) and of paper pulp was pasteurized at 70 °C for 24 h and dried overnight in a sterile flow cabinet. Each pasteurized dry material was then mixed under sterile conditions with NPK fertilizer (0.24 g fertilizer g^−1^ wood or paper pulp) and brought to a moisture content of 60% WHC. Portions containing ca. 4.2 g fertilized and pasteurized material were equally distributed in Petri dishes (⌀ 9 cm). For each material, five replicates were prepared. The ability to grow on the woody materials and paper pulp was tested for *R. solani* AG 2-2IIIB (IRS, Bergen op Zoom, The Netherlands), a strain that causes diseases in beetroot, sugar beet, and other vegetable crops [36]. The *Rhizoctonia* strain was pre-grown on potato dextrose agar (PDA, Oxoid, Badhoevedorp, The Netherlands) at 25 °C. Plugs of mycelium-covered agar (⌀ 3 mm) were taken with a cork borer and placed in the center on top of the substrate, with the mycelium in contact with the substrate material. For each material, three additional replicate Petri dishes (⌀ 5 cm, with 2.8 g of sawdust or paper pulp) were incubated without pathogen inoculation, as a control for the absence of other decomposer microbes. The Petri dishes were incubated in a dark climate chamber at 20 °C for 10 days.

### 2.3. RsP Experiment: Measurement of Growth Performance of R. solani

The growth performance of *R. solani* was evaluated by measuring the extension of hyphae (mycelial area) and ergosterol concentration in the substrate covered by the mycelium, as a proxy for mycelial density. After 10 days of growth, the area of mycelium of *R. solani* growing radially toward the edge was plotted on mica plastic sheets under a microscope (M250C, Leica Microsystems, Wetzlar, Germany) for each Petri dish. The plastic sheets were scanned, the areas of the mycelia were quantified using WinFOLIA software (Regent Instruments Inc., Ch Ste-Foy, QC, Canada). Next, the mycelium-covered portions of the substrates were harvested and transferred in 50 mL plastic vials. After freeze-drying and homogenization, a sample of 0.25 g of each substrate sample was stored in 4 mL of 10% KOH in methanol at −20 °C until ergosterol extraction.

The second experiment (Wood Types, WT) consisted of a bioassay with a soil that was naturally infected with *R. solani* to test fungus-stimulating and disease-suppressive effects of 10 cellulose-rich organic amendments: paper pulp, sawdust from beech, oak, hazel, elder, holly, willow, and cypress (Table 1); and 10% and 20% pre-decomposed beech sawdust. After sowing seeds of beetroot, germination and seedling performance followed.

### 2.4. WT Experiment: Sampling of Soil and Preparation of Materials

Soil was collected from an experimental field located near the village of Burgerbrug (52°75′96″ N, 4°71′07″ E, North-Holland, The Netherlands) in August 2018. In this field, different cultivars of beetroot (*Beta vulgaris*) were grown (Bejo Zaden, Warmenhuizen, The Netherlands). The soil had a loamy sand texture (4% coarse sand, 84% fine sand, 6% silt, 6% clay), pH 7.5, and organic matter 1%. Bulk soil from a depth of 0–10 cm was collected in between rows from four plots (4 × 2.5 m each) where beetroot plants showed the most severe signs of *Rhizoctonia* infection symptoms such as low germination rate, reduced plant biomass, dark root lesions, and yellowing of leaves. The soil sample was sieved (4 mm), homogenized with a 250 L mixer (Patriot 250, Atika, Burgau, Germany), and stored at 4 °C until use, for a maximum of two months.

Paper pulp and wood sawdusts were sourced as described in Section 2.1. Pre-decomposed beech sawdust was prepared, without soil, as follows. Beech sawdust and NPK fertilizer (Tuinmest 12-10-18, POKON Naturado, Veenendaal, The Netherlands) were mixed at a concentration of 0.24 g fertilizer g^−1^ wood to lower the C:N ratio of the material below 15:1, which prevents nitrogen immobilization by microbes and allows plant growth [36]. The wood–fertilizer mixture was brought to 60% water-holding capacity by adding deionized water, which was sterilized by autoclaving. Portions of the wood–fertilizer mixture corresponding to 80 g of dry sawdust were placed in eight replicate cylindric plastic jars (⌀ 10 cm × 17 cm), each closed with filter paper, and incubated at 20 °C in a dark climate chamber. After that, the decomposition of the mixture was determined weekly on the basis of weight loss of dry wood. Wood mass loss was 10% and 20% after 16 and 30 days, respectively. At each of these time points, four replicate jars were taken out from the climate chamber and stored at 4 °C until use, in order to slow down the decomposition process until use in the WT bioassay.

### 2.5. WT Experiment: Bioassay with Wood Sawdust Types and Paper Pulp

Sawdusts and paper pulp were added at a concentration of 5 g kg^−1^ dry soil and combined with NPK fertilizer. The control consisted of the addition of fertilizer only. NPK fertilizer (Tuinmest 12-10-18) was added at a concentration of 1.2 g kg^−1^ dry soil, corresponding to an input of 144 mg N, 120 mg P, and 168 mg kg^−1^ soil. Decomposed beech sawdust was added to the soil (5 g kg^−1^, both decomposed sawdust and soil calculated on dry weight basis) without additional NPK, as the NPK was already mixed with sawdust at the beginning of the decomposition period. The moisture of the amended soil was adjusted to 60% WHC, and the soil was transferred to pots (1.2 kg dry soil per pot). The experiment had five replicates for each amendment, and they were arranged in a random order within replicate blocks (CRBD). The amended pots were incubated for two weeks in the greenhouse under a dark cover, so that fungi present in soil would be stimulated before planting. After two weeks, each pot was sown with 32 seeds of a *Rhizoctonia*-sensitive beetroot cultivar (*Beta vulgaris* var. conditiva ‘Pablo’), at least 1 cm distance from each other. The plants were grown for three weeks before harvesting. Soil moisture was maintained constant on a weight basis.

### 2.6. WT Experiment: Determination of Germination Percentages and Plant Disease Incidence

The number of seedlings that emerged one and two weeks after sowing were counted. Single beetroot seeds of our stock could consist of clumped seeds and could give origin to either one, two, or seldomly, three seedlings. The percentage of emerged seedlings (G) was calculated as:(1)$$G\;=\left(C/\mathrm{Cmax}\right)\;\times \;100$$
where C is the number of emerged seedlings in a treatment and C_max_ is the number of germinated seeds under optimal conditions (disease-free potting soil). The number and health status of seedlings were monitored during the three-week growth period. As seedlings displayed stem and root lesions to various degrees, the severity of lesions was classified in five groups at the end of the growth period for each individual plant (shown in Figure 1). Classification criteria were: plants with little or no lesions in the crown area (D0); small brown lesions and/or a thinner diameter in the crown area (D1); brown or black lesions in the crown area, extending to the root and/or stem for up to 1 cm (D2); black lesions extending to most of the stem and/or roots (D3) and lesions extending to the leaves and involving the whole plant (D4). Several D4 plants died during the three-week growth period. The disease severity index (D) was calculated for each pot as follows, where C_DX_ indicates the count of seedlings in a pot for each disease class [37].
(2)D = (1×CD0+2×CD1+3×CD2+4×CD3+5×CD4)/(5×(CD0+CD1+CD2+CD3+CD4))×100

Plants belonging to the D0, D1, and D2 classes, namely with no or minor disease symptoms, were indicated as “successful” for a simplified display of the results, whereas plants classified as D3 and D4 were not likely to survive, therefore they were indicated as “unsuccessful”. The rate of successful emerged seedlings was calculated for each pot in the same way as the germination rate.

### 2.7. WT Experiment: Sampling of Soil and Plants

At the end of the three-week plant-growth period, the remaining plants in the pots were harvested. Roots and shoots were separated from each other, after classifying the disease symptoms (see Section 2.6). Shoots obtained from the same pot were pooled together and their biomass was measured after drying at 40 °C for 5 days.

One day before sowing and at the end of the plant-growth period, soil was sampled using a corer (⌀ 6 mm) in four random spots in each pot. The composite sample resulting from the four cores was homogenized and 1 g of it was stored in 4 mL of 10% KOH in methanol at −20 °C. This was used for ergosterol extraction within three months.

In the third experiment (time of sowing, ToS), a second batch of the *Rhizoctonia*-infected soil was amended with five organic materials, including two wood sawdust types, paper pulp, and two materials of animal origin (Table 1). Beetroot was sown at three time intervals after the amendment application, in order to test the effect of amendment timing on the disease suppression.

### 2.8. ToS Experiment: Bioassay with Varying Pre-Incubation Times of Soil Amendments

The soil was sampled in June 2019 from the same location and with the same method as indicated for the WT experiment in Section 2.4.

The ToS experiment included paper pulp and two woody materials (5 g kg^−1^ dry soil), namely beech sawdust and oak sawdust (sourced as indicated in Section 2.1), combined with NPK fertilizer (1.2 g kg^−1^), as well as two animal-derived materials (Table 1): pig hair meal (Darling Ingredients, Irving, TX, USA and Sonac Burgum, Sumar, The Netherlands) and shrimp meal (Telson, Lauwersoog, The Netherlands), added at lower concentrations (2 g kg^−1^ dry soil) and without NPK fertilizer. The lower concentration used for animal-derived materials without mineral fertilization was chosen to provide a N input comparable to the other soil treatments [36]. The control was amended with NPK fertilizer only. Amended soils were adjusted to 60% WHC and, for each treatment, 15 pots containing the equivalent of 1.3 kg soil (dw) were prepared, making the total 90 pots. Next, 32 beetroot seeds per pot were sown 1 day after amendment in five replicate pots for each soil treatment (T1). The other pots were incubated without plants under a dark cover. Of these, five replicates for each soil treatment were sown 14 days after amendment (T2), whereas the remaining five replicates per treatment were sown 28 days after amendment (T3, Figure 2). In all cases, plants were grown for three weeks after sowing, namely until day 21, day 35, and day 49, for T1, T2, and T3, respectively. All pots were kept in the greenhouse during the experiment, randomly arranged in five blocks (CRBD). The germination rate, the rate of emergence of healthy and heavily diseased plants, and the disease severity index were measured and calculated as described above for the WT experiment (see Section 2.7). Roots obtained from each pot were pooled, freeze-dried, grinded to a fine powder by beating with metal beads, and stored at room temperature before DNA extraction.

### 2.9. ToS Experiment: Sampling of Soil and Plants 

Soil was sampled from all the pots of the experiment non-destructively at days 1, 3, 7, 14, 21, 28, 35, 42, and 49 from the start of the experiment (Figure 2). At each sampling, a composite soil sample of about 5 g was obtained by taking four cores of soil (⌀ 6 mm) from random spots in a pot, without disturbing seeds or the plants. Part of the composite sample (1 g) was stored in 4 mL of 10% KOH in methanol at −20 °C for ergosterol extraction. The rest of the sample was freeze-dried and stored at room temperature until DNA extraction.

Three weeks after sowing, the seedlings were harvested. The disease level of seedlings was classified (see Section 2.6). Roots obtained from all the plants of same pot were rinsed with water, pooled together, freeze-dried, grinded by bead-beating with metal beads, and subsequently stored at room temperature. The roots included in the pooled root sample were obtained from plants of all levels of disease (D0–4).

### 2.10. RsP, WT, and ToS Experiments: Fungal Biomass

Fungal biomass in wood (RsP experiment) and soil (WT and ToS experiments) was estimated by the measurement of ergosterol concentration. For the ToS experiment, only a selection of samples was subjected to ergosterol extraction, in order to compare ergosterol- and qPCR-based estimation of total fungal abundance. Alkaline extraction of ergosterol was performed starting from 0.25 g wood samples and 1 g soil samples, as described in earlier studies [20]. Ergosterol concentration was then quantified by LC-MSMS (UHPLC 1290 Infinity II and 6460 Triple Quad LC-MS, Agilent Technologies, Santa Clara, CA, USA).

### 2.11. ToS Experiment: DNA Extraction, qPCR of R. solani and Fungi, and Sequencing of Fungal and Bacterial Communities

DNA extraction and qPCR were performed for soil and root samples from the ToS experiment. Soil samples taken at days 3, 7, 14, 21, 28, 35, 42, and 49 and root samples were included for control, and paper pulp and oak amendments. For beech, hair meal, and shrimp meal, samples of roots and soil were only taken at days 7, 21, 35, and 49, as these materials had less-evident effects on plant performance. DNA was extracted from 0.25 g soil or 0.25 g root with the DNeasy PowerSoil Pro Kit (Qiagen, Hilden, DE, USA) according to the manufacturer’s instructions. The abundance of *R. solani* was quantified by qPCR targeting a 174 bp fragment within the ITS region with primers ARSF5 (5′-ACTAAGTTTCAACAACGGAT-3′) and ARSR5 (5′-TTACTTTGAAGATTTCATGA-3′) [38], whereas the total fungal abundance was quantified by targeting the ITS2 region (350–750 bp) with primers ITS9f (5′-GAACGCAGCRAAIIGYGA-3′) and ITS4r (5′-TCCTCCGCTTATTGATATGC-3′) [39]. Samples were analyzed in two technical replicates, arranged in a blocked random order. The qPCR was performed with a CFX Connect Real-Time PCR Detection System (Bio-Rad Laboratories, Hercules, CA, USA) in 20 μL mixtures containing 10 μL SYBR Green PCR Master Mix (Sigma-Aldrich, St. Louis, MO, USA), 1 μL BSA 4 mg ml^−1^, 0.8 μL of each primer, 3.4 μL DEPC-treated water (Sigma-Aldrich, St. Louis, MO, USA), and 18–45 ng (4 μL) template DNA. The qPCR cycling conditions for *R. solani* were: denaturation at 95 °C for 5 min, 40 cycles of 95 °C for 10 s, 54 °C (*R. solani*) or 56 °C (ITS2) for 20 s, and 72 °C for 25 s, followed by a final extension step at 72 °C for 10 min. Plasmids containing DNA amplicon from *R. solani* AG2-2IIIB and *Trichoderma koningi* were used as a standard for the quantification of *R. solani* and ITS2 copy number, respectively.

Based on the results for plant performance and qPCR, a selection of DNA samples from the ToS experiment was subjected to ITS2 and 16S amplicon sequencing with the primer pairs ITS4r/9f [39] and Eub515f/806r [40], respectively. The sample selection included soil samples from days 1, 7, 21, 35, and 49, and root samples for experimental units belonging to control and paper pulp and oak sawdust treatments. The preparation of barcoded libraries and the sequencing by Illumina MiSeq PE250 were performed by McGill University and Génome Québec Innovation Centre, Montréal, QC, Canada.

### 2.12. Bioinformatic and Statistical Analyses

The statistical analysis was carried out in R (v 3.4.0) [41]. For the RsP experiment, differences in area and local ergosterol concentration of *R. solani* mycelia growing in 15 cellulose-rich substrates were analyzed using one-way ANOVA for each variable, followed by Tukey’s post hoc test for pairwise comparisons (5% family-wise error rate). Log-transformation was applied to both ergosterol and area data, in order to meet the assumptions of normality and equality of variances.

For the WT experiment, the effect of soil treatment on germination rate, number of healthy seedlings, total shoot biomass per pot, and disease severity index in each pot was analyzed with one-way ANOVA models, after checking the assumptions of normality and homogeneity of variance. One-way ANOVAs had soil treatment as an explanatory factor and a block as a random factor, and Dunnett’s post hoc test was used to compare the effect of each organic material to the control. For soil ergosterol concentration in the WT experiment, measured before and after plant growth, a repeated-measures ANOVA was used, with soil treatment as the between-subjects factor, time point as the within-subjects factor, and a block as a random factor. Differences in ergosterol concentration between each material and the control for each time point were analyzed based on pairwise comparisons with Dunnett’s post hoc test (family-wise error rate 5%). Pearson’s correlation index and its significance (*p* < 0.05) were calculated for the relation between soil ergosterol and disease severity index, as well as between soil ergosterol and the rate of successful plants. Disease severity and rate of successful plants were also correlated with the performance (mycelial area and density) of *R. solani*, measured in the RsP experiment (*p* < 0.05).

For the ToS experiment, the number of germinated plants, number of successful seedlings, and the disease severity were analyzed with two-way ANOVA models, with soil treatment and sowing time as fixed factors and a block as a random factor. Pairwise comparisons (Tukey’s post hoc test with 5% family-wise error rate) were used to compare each variable between each soil treatment and the control within each of three sowing times. Pairwise comparisons between treatments within each sowing time and between sowing times within control pots were checked as well. The qPCR-based abundance of fungi was analyzed using a generalized linear model with fitted gamma distribution of errors. Soil treatment, sowing time (T1–3) and day of sampling were used as predictive factors, whereas block and pot identity were random factors. Moreover, the effect of each soil treatment on fungal abundance, as compared to the control, was analyzed for soil sampled one week after sowing, at harvesting and for root samples, by combining two-way ANOVA (with soil treatment and sowing time as factors) and Dunnett’s post hoc test (5% family-wise error rate), both tests were applied on log-transformed data, in order to fit the assumption of normality and homoscedasticity. A generalized linear model was applied to qPCR-based abundance of *R. solani*, with Poisson distribution of errors, in order to account for a large number of zeros in the dataset. In this case, soil treatment, sowing time (T1–3) and day of sampling also were used as predictive factors, whereas block and pot identity were random factors. The *R. solani* qPCR-based abundance was log-transformed and analyzed for the treatment effects in soil one week after sowing and in root samples by using Welch’s test for two-way ANOVA combined with Dunnett’s post hoc test (5% family-wise error rate). Differences among soil treatment and sampling days were analyzed for qPCR- and ergosterol-based measurement of fungal abundance with two-way ANOVA combined with Tukey’s post hoc test.

The fungal sequencing data obtained for the ToS experiment was first processed with ITSxpress in order to extract the ITS2 region [42]. After that, the R package dada2 was used for quality filtering (maxEE = 2, truncQ = 2), for joining pair-end reads, removing chimeric sequences, modelling sequencing errors, and finally identifying sequence variants (SVs) with the DADA2 algorithm [43]. The UNITE v2019 database [44] was used for the taxonomical assignment of SVs with an RDP classifier. The bacterial sequencing data was directly processed in R with DADA2, following the same pipeline. The filtering parameters were: truncLen = 240, maxEE = 2, truncQ = 2; and the reference database for taxonomical identification of SVs was SILVA v132. The fungal and bacterial dataset counted 2,394,488 and 3,290,351 reads, respectively. Singletons and doubletons were removed, as well as SVs that were not assigned to fungi and bacteria (i.e., mitochondria, chloroplasts), resulting in 2418 fungal SVs and 18,746 bacterial SVs.

Permutational multivariate analysis (PERMANOVA, vegan) was used to determine the effect of organic amendments on bacterial and fungal community composition. Sowing time and day/compartment of sample (soil or root) were also included as fixed factors in the model and permutations were controlled by block (strata). PERMDISP (vegan) revealed a low homogeneity of dispersion in both the fungal and bacterial data sets. The relative abundance data were log + 1 transformed before performing PERMANOVA. The Bray–Curtis dissimilarity in fungal and bacterial community composition for paper pulp and oak relative to the control was quantified with the usedist package as the distance between the centroids of each group. The *R*^2^ and significance of such distances was tested with pairwise comparisons for multivariate data (pairwise.adonis, with FDR adjustment for multiple testing). Differential abundance analysis (DESeq2, Wald test) was performed for both fungal and bacterial families, in order to identify which were significantly affected by oak sawdust and paper pulp amendments, as compared to the control. SVs were aggregated at family level, zero-count families were removed, and a pseudocount was added in order to adjust the algorithm sensitivity to low-abundance groups [45]. Differential abundance of fungal and bacterial families were analyzed independently for each day of sampling and for root samples at each sowing time (T1–3). The relative abundance of *R. solani* (teleomorph: *Thanatephorus cucumeris*) found in the fungal sequencing data set was analyzed with a generalized linear model with normal distribution of errors with soil amendment, sowing time and day as fixed factors Block and pot were random factors. This was followed by Welch’s ANOVA and Dunnett’s post hoc test, performed independently for each day and for roots.

## 3. Results

### 3.1. RsP Experiment

The pasteurization of sawdust types and paper pulp was sufficient to inhibit the growth of the natural fungal inhabitants of these materials, as assessed by the absence of development of hyphae on non-inoculated materials. *R. solani* was able to grow on all cellulose-rich substrates in absence of competition with other fungi. The development of mycelium of *R. solani* varied with the type of material (ANOVA, *F*_14,75_ = 229.4 and *F*_14,75_ = 37.4 for areal expansion and density (ergosterol), respectively; both *p* < 0.001). The smallest mycelial areas of *R. solani* were seen on sawdust of conifer trees (Douglas fir and cypress) and walnut (Figure 3, Appendix A). Of the deciduous wood types, willow and elder gave the largest mycelial area, but with low (elder) or intermediate (willow) mycelial density (Figure 3, Appendix A). A good performance of *R. solani* (large mycelial area, high fungal density) was seen for hazel, black alder, and snowy mespilus. Dense growth but small areal coverage was obtained with oak, holly, and hawthorn (Figure 3, Appendix A). By far the largest biomass increase of *R. solani* was seen for paper pulp (Figure 3, Appendix A).

### 3.2. WT Experiment

In the WT experiment, the selected cellulose-rich amendments increased fungal biomass in the soil (Figure 4), with exception of cypress sawdust. The ergosterol-stimulating effect of the amendments was distinct at each time point (interaction treatment × time point: *F*_10,55_ = 6.97, *p* < 0.001). At the time of sowing beetroots, two weeks after amendment, the largest increase in ergosterol was seen for paper pulp, beech, and 10% and 20% pre-decomposed beech and hazel sawdust. Elder sawdust had no effect on fungal biomass at week 2, but had increased ergosterol content at week 5, corresponding to the end of the plant-growth period. All other deciduous wood types, as well as paper pulp, had a higher ergosterol content as compared to the control at both week 5 and week 2. However, ergosterol concentrations at week 5 were either similar (oak, holly) or lower as compared to week 2 (fresh and pre-decomposed beech sawdust, hazel, willow, and paper pulp).

The average number of germinated beetroot seeds was higher for all amendments than for the control, although the overall effect was not significant (*F*_10,55_ = 1.62, *p* = 0.14). The strongest stimulation of germination was seen for 20% pre-decomposed beech sawdust, elder sawdust and paper pulp (Figure 5A). Soil organic amendments significantly affected the development of successful seedlings (*F*_10,55_ = 7.28, *p* < 0.001; Figure 5B). In particular, addition of paper pulp and oak and elder sawdust resulted in more successful seedlings than the control. These amendments also supported a significantly higher aboveground biomass of beetroot seedlings (Figure 5D). Disease severity indices were lower for elder and paper pulp amendments as compared to the control, whereas 10% pre-decomposed beech sawdust showed a slight increase of the disease severity index (Figure 5C). Both disease severity and number of successful seedlings were not significantly correlated with the soil ergosterol level at the time of sowing (*p* = 0.3, *R* = 0.16 and *p* = 0.2, *R* = 0.15, respectively) or the in vitro performance of *R. solani* on pasteurized materials (*p* = 0.8, *R* = −0.03 and *p* = 0.2, *R* = 0.21, respectively).

### 3.3. ToS Experiment

#### 3.3.1. Effect of Organic Amendments on Plants, Total Fungal Biomass, and *R. solani*

In the ToS bioassay, organic amendments had a significant impact on the total germination rate of beetroot seeds as compared to the control (*F*_5,90_ = 10.31, *p* < 0.001). However, the effect was dependent on the sowing time (*F*_10,90_ = 1.83, *p* = 0.08). For T1, only hair meal resulted in an increased germination rate (*p* < 0.05), whereas at T2 a higher number of germinated seedlings was seen for all soil amendments but shrimp meal (Figure 6A). A higher germination rate as compared to the control was also seen in pots sown one month after mixing (T3) for all amendments, with the exception of hair meal (Figure 6A). In the control treatments, the average number of germinated seedlings was highest at T1, albeit not significantly.

Both organic amendment and sowing time affected the amount of seeds resulting in successful plants (*F*_10,90_ = 3.5, *p* < 0.001). Oak sawdust amendment gave a higher number of successful seedlings at T1, T2, and T3 as compared to the control (Figure 6B). An increased number of successful seedlings was seen for beech sawdust only at T3 (*p* < 0.05), whereas for hair meal only at T1 (*p* < 0.1). Paper pulp had a low number of successful plants at T1, albeit not significant different from the control. At the longer time intervals between amendment and sowing, paper pulp gave a significantly higher number of successful plants (T2: *p* < 0.1; and T3: *p* < 0.05), as compared to the control. In control pots, the number of successful plants increased, albeit not significantly, with longer time gaps between start of incubation and sowing (T1–3; Figure 6B).

Organic amendments had an impact on disease severity only at T1 (*F*_10,90_ = 8.6, *p* < 0.001; Figure 6C). Most prominent was the effect by oak (*p* < 0.001). Beech sawdust caused only a slight decrease in disease severity as compared to the control at T1 (*p* < 0.1). An opposite pattern was seen for paper pulp at T1, with an increased disease severity (*p* < 0.001). At T2 and T3, organic amendments had no significant effect on disease severity. The disease severity index in the control decreased with increasing incubation times before sowing (Figure 6C).

Soil amendments affected the total fungal abundance in soil and roots. The effects varied with the sampling day after amendment and, for same sampling days, with amendment–sowing time-intervals (GLM, interactive term treatment × ToS × day *F*_10,480_ = 3.50, *p* < 0.001; Appendix A). During seedling germination (one week after sowing), the copy number of total fungi was increased in soil treated with oak, beech, and paper pulp, as compared to the control, independently of the time interval between amendment and sowing. At the end of the plant-growth period, the fungal copy number was still higher as compared to the control for oak- and beech-amended soil, whereas paper pulp-, hair meal-, and shrimp meal-amended soil had lower fungal copy numbers (Figure 7, Appendix A). Fungal colonization of roots at the end of the growth period was comparable for plants grown in control, oak-, beech-, and paper-pulp-amended soil, while hair meal (T1) and shrimp meal (T1 and T2) amendments resulted in lower fungal abundance in plant roots (Figure 7, Appendix A).

Since different methods for fungal biomass quantification were used (qPCR in ToS and ergosterol in WT), a comparison was made of the two methods for soils of the WT control, oak sawdust, and paper pulp amendments. This revealed similar trends of fungal abundances in the different treatments (Appendix A).

A simple effect of soil amendments was seen for the abundance of *R. solani* in soil and roots (GLM, treatment *F*_5,480_ = 19.1, *p* < 0.001; Appendix A). In particular, higher copy numbers of *R. solani* were detected in soil amended with paper pulp, compared to the control, during the first week after amendment (Figure 8A). Roots of seedlings sown at T1 also contained higher *R. solani* numbers in the paper pulp-amended soil (Appendix A). For T2, plants grown in soil amended with beech, oak, hair meal, and shrimp meal had a lower abundance of *R. solani* in roots as compared to the control (Figure 8A, Appendix A). Similarly to the qPCR-based quantification of *R. solani* abundance, the relative abundances of *R. solani*, detected via ITS2 amplicon sequencing, increased in response to paper pulp amendment (Figure 8B; GLM *R*^2^ = 0.62, *χ*^2^ = 0.25, *df* = 38). Such a change was seen in paper-pulp-amended soil at day 7, irrespectively of the sowing time (Welch’s ANOVA *F*_2,45_ = 10.7, *p* < 0.05; Dunnett’s, *p* < 0.01), whereas the effect of paper pulp amendment on *R. solani* was not significant at days 21, 35, and 49. Paper pulp changed the proportion of *R. solani* in all plant roots (Welch’s ANOVA *F*_4,45_ = 5.89, Dunnett’s *p* < 0.01 for T1, *p* < 0.05 for T2 and T3), even though plants sown at T2 and T3 had an overall lower proportion of *R. solani* as compared to T1.

#### 3.3.2. Effect of Oak Sawdust and Paper Pulp on Fungal and Bacterial Communities

Oak sawdust and paper pulp amendments strongly affected the composition of both fungal (*R*^2^ = 0.3, *p* < 0.001; Appendix A) and bacterial communities (*R*^2^ = 0.02, *p* < 0.001; Appendix A) in soil and in roots. The effect of these amendments varied depending on the day of sampling and compartment (soil/root) for both fungi (*R*^2^ = 0.3, *p* < 0.001; Appendix A) and bacteria (*R*^2^ = 0.08, *p* < 0.001; Appendix A). Oak sawdust caused an immediate, large, and persistent shift in the soil fungal community relative to the control soil and roots (Table 2A, Appendix A), and a smaller effect in the soil bacterial community (Table 2B, Appendix A). For fungi, the changes were ascribed to a higher proportion of Sordariomycetes, in particular of *Lasiosphaeriaceae* and *Chaetomiaceae* (Appendix A; Wald *p* < 0.01). Oak sawdust also increased the relative abundance of several fungi belonging to Sordariomycetes (T1–3), Dothideomycetes, *Olpidiaceae*, and *Orbiliaceae* (T1 and T2) in root tissues (Appendix A; Wald *p* < 0.05). For bacteria, an increase was seen in the proportion of Bacteroidia (*Flavobacteriaceae*), α-Proteobacteria (*Rhizobiaceae* and *Sphingomonadaceae*), and γ-Proteobacteria (*Cellvibrionaceae* and *Methylophyliaceae* (Figure 9A); Wald *p* < 0.01) in soil in response to oak sawdust, which was most pronounced at day 7.

For paper pulp, a strong, yet short-lived, shift was seen in the bacterial community composition in soil as compared to the control (Table 2B, Appendix A). Changes in the fungal community composition were smaller for paper pulp amendment than for those observed for oak sawdust (Table 2A, Appendix A). In particular, members of Orbiliomycetes and Agaricomycetes (including *R. solani* (Figure 8A) and unassigned Agaricomycetes) and *Chaetomiaceae* had a higher abundance in soil after paper pulp amendment at sampling day 7 (Appendix A; Wald *p* < 0.01). In roots, paper pulp increased the proportion of *Chaetomiaceae*, other Sordariomycetes, and Orbiliomycetes for plants of all sowing times, whereas Saccharomycetes, Mortierellomycetes, and Agaricomycetes (including *R. solani*) had a higher proportion in roots only for T1 (Appendix A). Paper pulp increased the proportion of Bacteroidia (*Cytophagaceae*, *Flavobacteriaceae*), *Cellulomonadaceae*, α-Proteobacteria, and γ-Proteobacteria (*Cellvibrionaceae* and *Legionellaceae* (Figure 9B); Wald *p* < 0.001 (Appendix A)). Such changes in the bacterial community composition were observed at day 7 and were less pronounced at days 21, 35, and 49 (Table 2B, Appendix A). *Flavobacteriaceae*; members of α-, γ-, and δ-Proteobacteria; Bacilli; and Verrucomicrobia were detected in higher proportions in the roots of seedlings grown in paper-pulp-amended soil for T2 and T3; whereas for T1, the effect of paper pulp was limited to an increased proportion of *Flavobacteriaceae*, among root-associated bacteria (Appendix A; Wald *p* < 0.05).

## 4. Discussion

This study showed that cellulose-rich amendments influenced the soil disease suppression in *R. solani*-infected arable soil. The effect of such materials on the performance of plants depended on their ability to activate soil-borne saprotrophic fungi, bacteria, and *R. solani* during the first weeks after amendment.

### 4.1. Growth of R. solani on Woody Substrates and Paper Pulp

Among the tested pasteurized, cellulose-rich materials, paper pulp was the most suitable substrate for the growth of *R. solani*. Paper pulp is constituted mainly of cellulose and is virtually devoid of lignin and hemicelluloses [46]. Therefore, the result was in line with previous findings of a good ability of *R. solani* to utilize pure cellulose for growth [23,25]. Indeed, *R. solani* is known to produce plant cell-wall degrading enzymes, including cellulases [30,31], which enable it to grow both as necrotroph and saprotroph [25,32]. The lower performance of *R. solani* on wood sawdusts can be attributed to partial shielding of cellulose fibers by lignin. Although *R. solani* has been reported to produce ligninolytic enzymes [31,47], their ability to modify lignin in woody substrates is unclear. In addition to this, a possible activity of ligninolytic enzymes could have been inhibited by high N concentrations, provided in this study in the form of NPK fertilizer mixed with wood sawdust, both in Petri dishes and soil [48,49]. Among wood types, the performance of *R. solani* on coniferous wood (Douglas fir and cypress) was much lower than on most deciduous wood species. Conifer wood possesses more recalcitrant lignin as compared to deciduous tree species, given by a higher content in guaiacyl units and higher degree of crosslinks [50,51]. Moreover, the biodegradability of wood is dependent on the composition and concentration of non-structural metabolites like terpenes, alkaloids, and phenolics. High amounts of diterpenes and lignans, two of the major components of resins, are found in conifer wood and act as fungistatics or fungicides [52,53,54]. In particular, wood from *Cupressaceae* species contains tropolones, which are among the strongest fungitoxic wood extractives [52,54].

Among deciduous tree species, a low performance of *R. solani* was found on sawdust of walnut wood, which previously had been reported to be resistant to degradation due to the presence of gallic acid, 2.7-dimethylphenantheren and juglone [52,55]. *R. solani* performance on the other deciduous tree species was overall higher as compared to conifer and walnut wood, but variable across tree species, both in terms of mycelial extension and density. This can be ascribed to differences in the amounts of biodegradable components, such as cellulose, hemicellulose and non-toxic extractives, which have a large variation in angiosperm wood [54,56]. Wood of birch, beech, maple and willow contain mostly decomposable extractives, such as simple phenolics, phenolic glycosides, fats and steroids [57], which makes them not particularly resistant to degradation [52,57,58]. Oak wood is characterized by the presence of hydrolyzable tannins [52,57], whereas elder wood contains cyanogenic glycosides and lectins [59,60]. These compounds are moderately toxic to some decay fungi [61,62,63,64] and possibly altered the growth of *R. solani* in this study, which developed relatively small or thin mycelia on oak and elder sawdust, respectively.

### 4.2. Effect of Wood Sawdusts and Paper Pulp on Fungal Biomass and Beetroot Seedling Performance in R. solani-Infected Soil

In the first bioassay with an *R. solani*-infected soil, clear positive effects on the number of healthy seedlings were seen for paper pulp, oak sawdust and elder sawdust, whereas effects of holly and beech sawdust were smaller. Sawdusts from willow, hazelnut and cypress had no significant effect on the seedlings. Positive effects of woody materials and paper pulp on *R. solani* disease suppression were reported in previous studies, although these materials have also been associated with negative effects on plant performance due to N immobilization [23,26]. In the current study, extra fertilization was applied and was sufficient for compensating the temporary N incorporation caused by the saprotrophic growth of fungi and bacteria.

Most deciduous wood sawdust amendments stimulated fungal biomass in the two weeks before the seeds were sown. However, in contrast to our expectation, the extent of total fungal biomass stimulation in soil by sawdust types did not explain the differences in their effect on the protection of seedlings against damping-off disease. This expectation was based on the possibility of increasing competitive and antagonistic interactions against pathogenic fungi, as a consequence of stimulation of saprotrophic fungal biomass and activity in arable soil [16,20,65]. In a previous study, it was shown that the community composition of decomposer fungi stimulated by sawdust in soil can differ among deciduous tree species [20]. Hence, wood chemistry likely influenced the composition and activity of decomposer fungi, which, in turn, may have determined the degree of disease suppression. Presence of fungistatic compounds in certain sawdusts may have selected tolerant decomposer species, and the release of such compounds by their decomposing activities may have contributed to inhibition of the growth of *R. solani* [64,66,67,68]. This may explain why oak and elder sawdusts gave a high suppression of *R. solani* damping-off, although the increase of total fungal biomass by the sawdust was relatively low.

The coniferous sawdust (cypress) used in this bioassay did not stimulate soil fungi or affect the health of beetroot seedlings. In an earlier study, a low response of soil fungi was also seen after amendment with another coniferous sawdust (Douglas fir) [20]. As discussed in Section 4.1, this may be due to composition and arrangement of lignin in coniferous wood. However, previous research showed that lignin extracted from conifer wood can reduce the viability of *R. solani* in soil due to the damage on its sclerotia by the oxidative action of ligninolytic enzymes (manganese peroxidases) of other soil fungi [69]. It may be that longer incubation times of coniferous sawdust in soil are needed to obtain a similar effect.

Pre-decomposed beech wood had similar effects on fungal biomass stimulation as fresh beech sawdust, although part of the stimulated fungi were probably introduced with the pre-decomposed material. However, the effect of pre-decomposed sawdust on suppressing damping-off disease was lower than that of fresh sawdust. It may be that the fungi introduced with decomposed sawdust were already less active than the ones established on fresh sawdust, which resulted in a decreased competitive interaction with pathogens for root exudates.

Contrary to what was hypothesized, the performance of *R. solani* in pasteurized cellulose-rich materials, as tested in the Petri dishes, was not predictive for the disease suppressive effects of the same materials in *R. solani*-infected soil. For instance, *R. solani* had a moderately low in vitro growth on oak and elder sawdust, whereas it grew extensively on paper pulp. Despite such differences in supporting *R. solani* growth in vitro, all these materials had a positive effect on beetroot seedling performance. The apparent contradiction between strong in vitro growth support of *R. solani* by paper pulp and inhibition of *R. solani* damping-off disease of beetroot seedlings by paper pulp amendment prompted further examination on the effect of timing between sowing and amendment in more detail.

### 4.3. Impact of Timing of Organic Amendments and Sowing on R. solani Population and Disease Dynamics, and on Fungal and Bacterial Communities

In accordance with the hypothesis, paper pulp caused a transient increase in the *R. solani* population in soil. This can be explained as a stimulation of the saprotrophic activity of *R. solani*, which was supported by strong in vitro growth of *R. solani* on pasteurized paper pulp. This coincided with an increased soil conductivity of the pathogen, as higher *R. solani* copy numbers were found in the soil one week after amendment, and the disease could spread among seedlings germinating in the same time frame. However, the increase in *R. solani* numbers in soil enriched with paper pulp was relatively short-lived (<2 weeks). In fact, pre-incubation of amended soil for two weeks before sowing was sufficient to observe a decline in *R. solani* abundance and to obtain a positive effect of paper pulp on seedling performance. The latter was seen in both the WT and ToS experiments. Similarly to our results, Bonanomi and colleagues [23] observed that suppression of *R. solani* increased with time of cellulose decomposition in soil. In addition, Croteau and Zibilske [26] reported on a transient increase in the concentration of *R. solani* propagules in soil after paper mill amendment. In both studies, the transient disease-conduciveness stimulation by cellulose had a total duration of at least four weeks. In the current study, the faster decline of the *R. solani* population and simultaneous increase of beetroot seedling performance can be ascribed to the additional source of mineral nutrients. This does not only prevent the detrimental effects of nutrient immobilization on plant growth, but also causes a more rapid decomposition of cellulose and succession of decomposers [20,70].

A rise and decline of *R. solani* abundance in paper-pulp-amended soil was confirmed by both qPCR and sequencing data. The containment of a further spread of *R. solani* in paper-pulp-amended soil can be ascribed to the competitive action of other cellulose-degrading fungi and bacteria. Indeed, the rise in *R. solani* was accompanied by a higher proportion of cellulolytic fungi (Orbiliomycetes, *Chaetomiaceae*) [71,72] and bacteria (*Cellulomonadaceae*, *Cellvibrionaceae*, *Cytophagaceae*, *Flavobacteriaceae*) [73,74,75], as well as bacteria commonly found in decomposing plant litter (α-Proteobacteria) [76,77]. In particular, *Cytophagaceae* and *Flavobacteriaceae* are often found in soil and roots invaded by *R. solani*, and have been indicated to act as antagonist against *R. solani* [27,78,79,80].

Stimulation of *R. solani* in soil was not observed for oak or beech sawdust amendments, despite the ability to grow in vitro on these substrates in the absence of competitors. However, the exploitation of cellulose in woody material by *R. solani* was apparently not sufficient to compete successfully with saprotrophic soil fungi [81]. This suggests that fungi commonly found in arable soil [20,82,83], such as *Lasiosphaeriaceae*, *Chaetomiaceae* and *Halosphaeriaceae*, can effectively utilize wood sawdust, while in the meantime they outcompete the growth of pathogenic fungi, associate with roots, and stimulate plant performance overall. Although the effect of sawdust addition on pathogen dynamics could be different in other soils, the reported negative effects of sawdust on plant performance could be mainly due to the immobilization of nutrients [22].

In comparison with the first bioassay (Section 4.2), the second bioassay experiment had different results in terms of fungal biomass stimulation, especially with paper pulp. Such a difference could be explained by changes in the fungal community composition or activity between the two soil batches, which were sampled one year apart. In addition to this, reduction in pathogen pressure occurred during the second bioassay. A gradual decline in *R. solani* propagule concentration in control pots was also observed in a pot experiment by Croteau and Zibilske [26]. *R. solani* abundance and activity is highly dynamic in soils, thus it can be influenced by changes in abiotic and biotic conditions when transferred from the field to the greenhouse [82].

The presence of plants stimulated saprotrophic fungi in the control, but such stimulation was not seen with hair meal or shrimp meal amendments. Similarly, no stimulation of *R. solani* was seen with these keratin- and chitin-rich amendments. Hence, the observed positive effects on beetroot seedling performance by hair meal and shrimp meal amendments can be mainly ascribed to the stimulation of bacterial groups. Indeed, it has been shown that disease-suppressive effects of keratin and chitin are associated with an increased abundance antagonistic bacterial groups belonging to *Oxalobacteriaceae* and Bacteroidetes [83,84].

## 5. Conclusions and Perspectives

This study supports the idea that stimulation of saprotrophic fungi by cellulose-rich organic amendments has the potential to enhance natural biocontrol of the notorious soil-borne pathogen *R. solani*. However, the application of such amendments in practice should be carefully considered. Transient stimulation of *R. solani* by paper pulp coincided with transition from disease-conducive to disease-suppressive effects on beetroot seedling damping-off. In order to benefit from positive effects of paper pulp, it is essential to have a period of weeks of pre-decomposition in the soil before sowing. In this period, development of competitive and antagonistic fungi and bacteria can proceed. For sawdust, the effect on plant performance was dependent on the type of wood used. Hence, additional information is needed on the role of wood chemistry in suppression of pathogens and/or stimulation of antagonistic activities of microbes in arable soils. A closer examination of beech and oak sawdust amendments indicated that there was no stimulation of *R. solani* in soil, implying that the risk for enhancing disease by woody amendments is low.

## Figures and Tables

**Figure 1 microorganisms-09-01285-f001:**
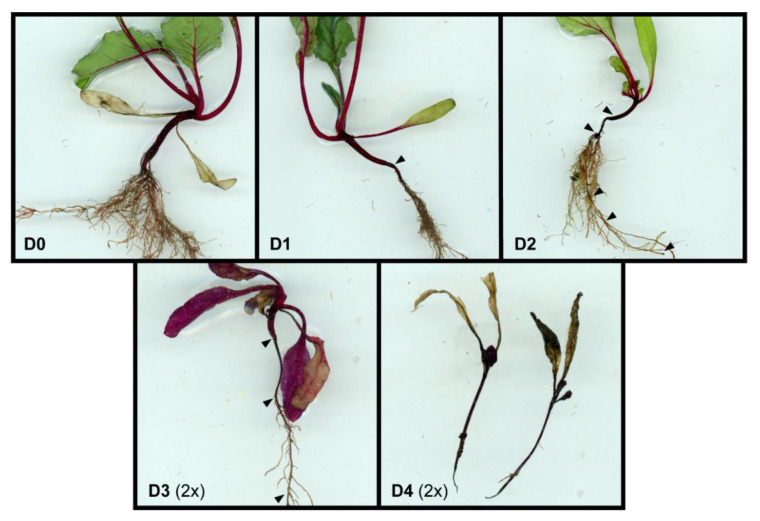
Classification of disease severity of beetroot seedlings based on lesions. Arrows indicate areas with dark lesions and thinning of the crown, stem, and root. Photos for D3 and D4 are shown at 2× magnification as compared to D0, D1, and D2. *R. solani* isolates were obtained from plants that had lesions, and they were identified as such by Sanger sequencing (data not shown). Photographs taken by the authors.

**Figure 2 microorganisms-09-01285-f002:**
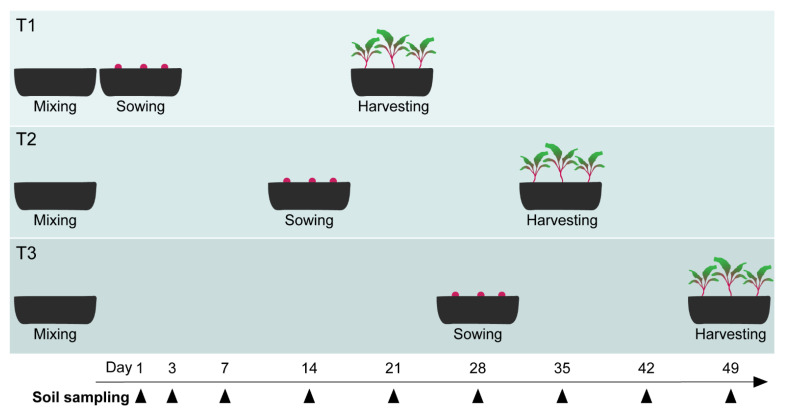
Experimental design of the ToS experiment. For T1, T2, and T3, the time gap in between soil organic amendments (day 0) and sowing is shown. The plant-growth period was the same for T1, T2, and T3 (3 weeks). Black arrows indicate the days when soil was sampled non-destructively from all pots. Additionally, at harvesting, root and shoot parts were collected. All pots were simultaneously incubated in a greenhouse.

**Figure 3 microorganisms-09-01285-f003:**
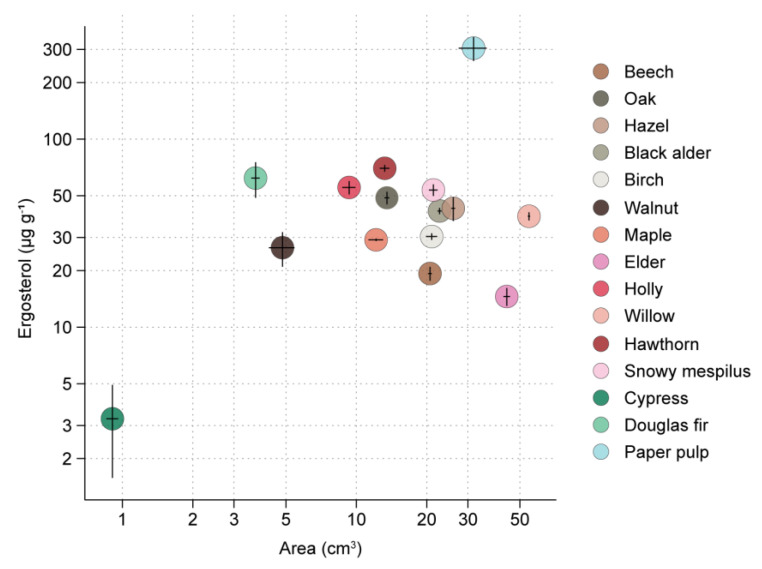
Performance of *R. solani* on 14 types of wood sawdust and paper pulp. The area of the substrates colonized by the mycelium of *R. solani* is shown on the *x* axis (mean ± SE, *n* = 5), whereas the *y* axis shows the ergosterol concentration of the substrate in the zone covered by *Rhizoctonia* hyphae, as a proxy for mycelial density (mean ± SE, *n* = 5). Both axes are log-scaled.

**Figure 4 microorganisms-09-01285-f004:**
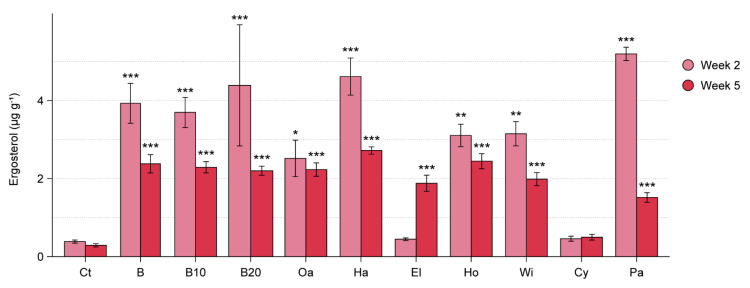
Fungal biomass stimulation in soil amended with sawdust of seven wood types, pre-decomposed beech sawdust, and paper pulp (WT experiment). Soil treatments are indicated as: Ct = control, B = beech sawdust, B10 = 10% pre-decomposed beech, B20 = 20% pre-decomposed beech, Oa = oak, Ha = hazel, El = elder, Ho = holly, Wi = willow, Cy = cypress, Pa = paper pulp. Ergosterol concentration is shown for amended and control pots (mean ± SE, *n* = 5) at the time of beetroot sowing (two weeks after amendment) and at harvesting of beetroot seedlings (five weeks after amendment). Significant differences between each treatment and the control (Dunnett’s test) are indicated as * 0.05 > *p* > 0.01; ** 0.01 > *p* > 0.001; *** *p* < 0.001.

**Figure 5 microorganisms-09-01285-f005:**
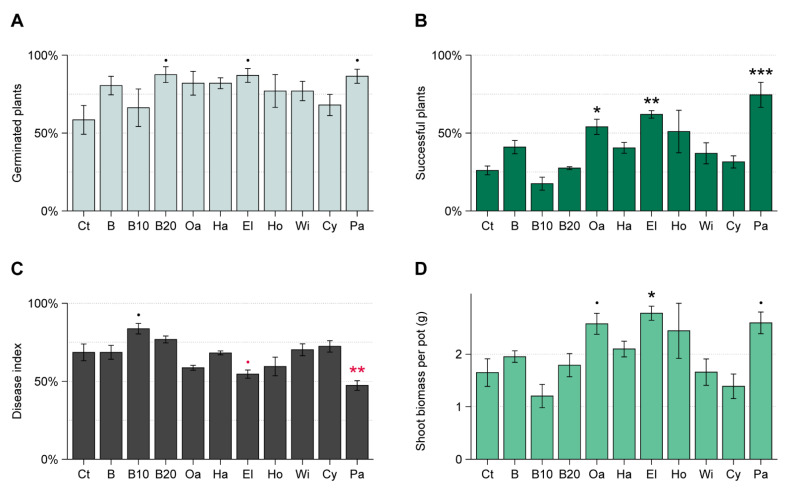
Performance of beetroot seedlings in *Rhizoctonia*-infected soil amended with sawdust types and paper pulp (WT experiment). Soil treatments are indicated as: Ct = control, B = beech sawdust, B10 = 10% pre-decomposed beech, B20 = 20% pre-decomposed beech, Oa = oak, Ha = hazel, El = elder, Ho = holly, Wi = willow, Cy = cypress, Pa = paper pulp. (**A**) Percentage of germinated seeds; (**B**) percentage of seeds that resulted in successful plants; (**C**) seedling disease severity index per pot; (**D**) total aboveground biomass per pot. All measurements are shown as mean ± SE, *n* = 5 for each treatment. Significant differences between each treatment and the control (Dunnett’s test) are indicated as • 0.1 > *p* > 0.05; * 0.05 > *p* > 0.01; ** 0.01 > *p* > 0.001, *** *p* < 0.001. Black and red symbols indicate significant increase and decrease, respectively, as compared to the control.

**Figure 6 microorganisms-09-01285-f006:**
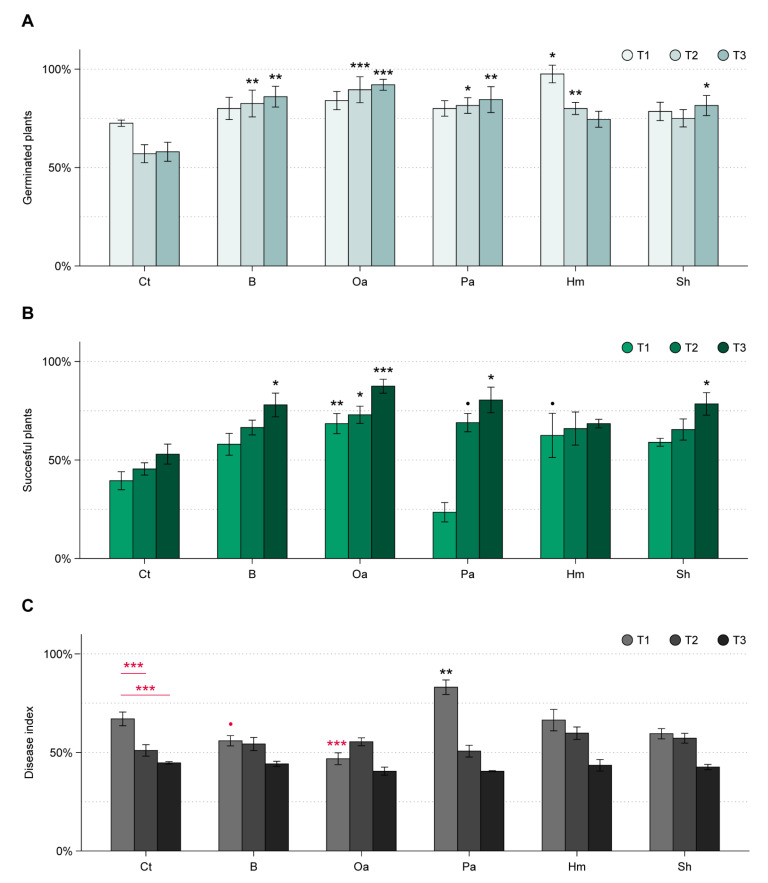
Performance of beetroot seedlings in soil amended with five organic materials, with three time intervals (T1 = 1 day, T2 = 14 days, T3 = 28 days) between amendment and sowing (ToS experiment). Soil treatments are indicated as: Ct = control, B = beech sawdust, Oa = oak sawdust, Pa = paper pulp, Hm = hair meal, Sh = shrimp meal. (**A**) Percentage of germinated seeds and (**B**) percentage of seeds that resulted in successful plants. (**C**) Disease severity index. All measurements are shown as mean ± SE, *n* = 5 for each treatment and sowing time. Significant differences between each treatment and the control within each time interval (Dunnett’s test), and among time intervals for the control (Dunnett’s test), are indicated as • 0.1 > *p* > 0.05; * 0.05 > *p* > 0.01; ** 0.01 > *p* > 0.001, *** *p* < 0.001. Significant increases and decreases are shown with black and red symbols, respectively.

**Figure 7 microorganisms-09-01285-f007:**
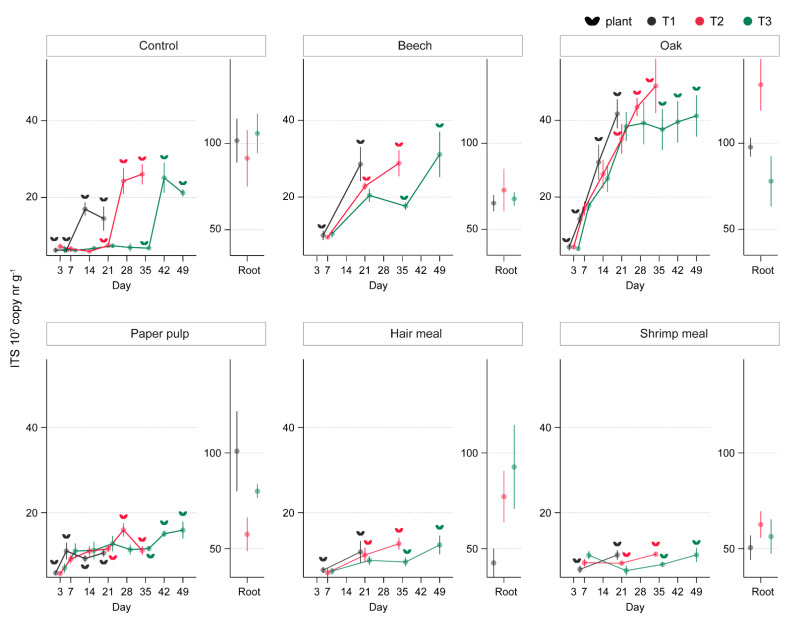
Quantification of fungal (ITS2) copy numbers in soil amended with five organic materials and sown at three time points after amendment. Fungal copy number was determined in soil of both planted and unplanted pots 1, 3, 7, 14, 21, 28, 35, 42, and 49 days after amendment (*x*-axis) for control, oak, and paper pulp; whereas for beech, hair meal, and shrimp meal, data were collected for days 7, 21, 35, and 49 (*x*-axis). For all soil treatments, fungal copy numbers in the roots are shown as mean ± SE, *n* = 5. Plant symbols next to each data point indicate that soil samples were obtained from planted pots.

**Figure 8 microorganisms-09-01285-f008:**
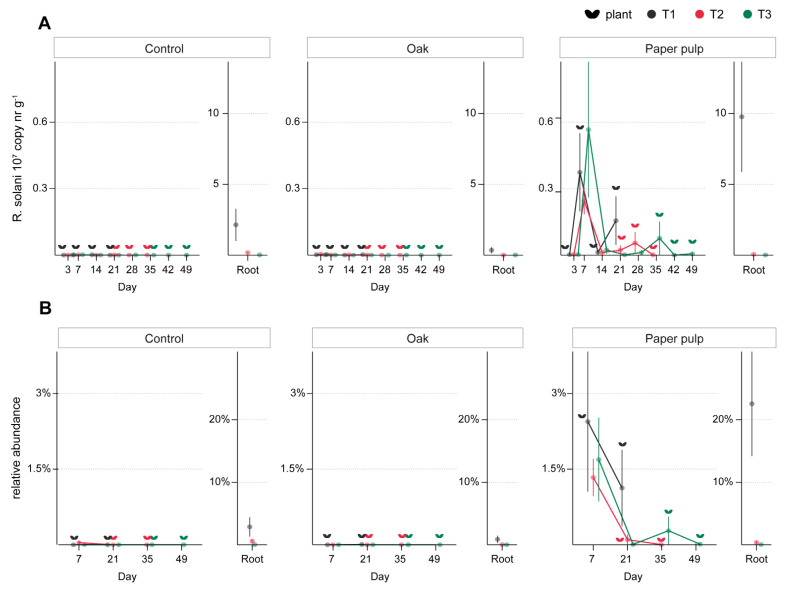
*R. solani* abundance in soil amended with oak sawdust and paper pulp and sown at three time points after amendment. (**A**) *R. solani* copy number was determined with qPCR in soil of both planted and unplanted pots at 1, 3, 7, 14, 21, 28, 35, 42, and 49 days after amendment (*x*-axis) and in roots (as mean ± SE, *n* = 5). (**B**) *R. solani* relative abundance was quantified by ITS amplicon sequencing in soil at 7, 21, 35, and 49 after amendment (*x*-axis) and in roots (as mean ± SE, *n* = 5). Plant symbols next to each data point indicate that soil samples were obtained from planted pots.

**Figure 9 microorganisms-09-01285-f009:**
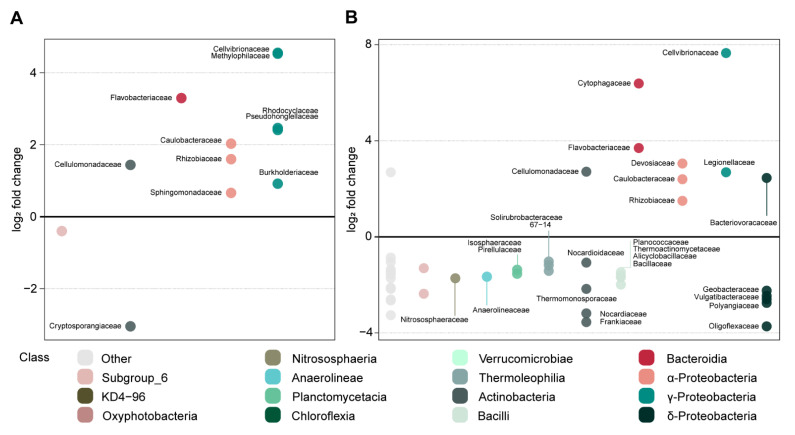
Effect of oak (**A**) and paper pulp (**B**) amendment as compared to control on bacterial families in soil sampled 7 days after amendment. Bacterial families significantly over- (log_2_ fold change > 0) and under-represented (log_2_ fold change < 0) were highlighted by differential abundance analyses (Wald test *p* < 0.01). Bacterial/fungal families are grouped by class along the *x*-axis; classes with relative abundance < 3% are indicated as “Other”.

**Table 1 microorganisms-09-01285-t001:** Wood types and organic materials used in this study in the three experiments: RsP (*R. solani* performance on organic materials in a Petri dish assay), WT (bioassay with soil amended with distinct wood types), and ToS (bioassay investigating the effect of organic amendments and the use of three times of sowing after amendment). Materials utilized in each experiment are indicated by the symbol •.

Material	Species	Exp. RsP	Exp. WT	Exp. ToS
Beech	*Fagus sylvatica*	•	•	•
10% decomposed beech	*Fagus sylvatica*		•	
20% decomposed beech	*Fagus sylvatica*		•	
Oak	*Quercus robur*	•	•	•
Hazel	*Corylus avellana*	•	•	
Black alder	*Alnus glutinosa*	•		
Birch	*Betula* sp.	•		
Walnut	*Juglans* sp.	•		
Maple	*Acer* sp.	•		
Elder	*Sambucus* sp.	•	•	
Holly	*Ilex* sp.	•	•	
Willow	*Salix alba*	•	•	
Hawthorn	*Crataegus* sp.	•		
Snowy mespilus	*Amelanchier* sp.	•		
Cypress	*Cupressus sempervirens*	•	•	
Douglas fir	*Pseudotsuga menziesii*	•		
Paper pulp	-	•	•	•
Hair meal	*Sus scrofa*			•
Shrimp meal	*Crangon crangon*			•

**Table 2 microorganisms-09-01285-t002:** Effect of paper pulp and oak amendment over time on fungal (A) and bacterial (B) community composition, relative to the control soil and roots. For each comparison, the Bray–Curtis dissimilarity distance between group centroids is indicated (Distance), as well as the proportion of variance explained by soil amendment (R2) and its significance (pairwise ADONIS, FDR-adjusted *p*-values: ** *p* < 0.01, * 0.01 < *p* < 0.05, • 0.05 < *p* < 0.1).

**A**															
**Comparison**	**Day 7**			**Day 21**			**Day 35**			**Day 49**			**Root**		
	Distance	*R* ^2^		Distance	*R* ^2^		Distance	*R^2^*		Distance	*R* ^2^		Distance	*R* ^2^	
Oak–control	0.55	0.68	**	0.68	0.73	**	0.70	0.79	**	0.69	0.72	*	0.32	0.21	**
Paper pulp–control	0.50	0.56	**	0.40	0.42	**	0.34	0.34	**	0.36	0.30	*	0.57	0.40	**
**B**															
**Comparison**	**Day 7**			**Day 21**			**Day 35**			**Day 49**			**Root**		
	Distance	*R* ^2^		Distance	*R* ^2^		Distance	*R^2^*		Distance	*R* ^2^		Distance	*R* ^2^	
Oak–control	0.27	0.05	*	0.25	0.04	•	0.30	0.06		0.37	0.10		0.26	0.04	
Paper pulp–control	0.43	0.12	*	0.30	0.06	*	0.33	0.07	*	0.47	0.16	•	0.33	0.06	*

## Data Availability

The data presented in this study are openly available in the Dryad repository at doi:10.5061/dryad.fqz612js6, and in the European Nucleotide Archive with reference number PRJEB40211 (ERP123824).

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
