# Peer review of "Impact of Cellulose-Rich Organic Soil Amendments on Growth Dynamics and Pathogenicity of Rhizoctonia solani"

_microorganisms, 2021, doi:10.3390/microorganisms9061285_

Round 1

Reviewer 1 Report

 In the manuscript ‘Impact of cellulose-rich organic soil amendments on growth dynamics and pathogenicity of Rhizoctonia solani’ Clocchiatti and collaborators investigated the potential of fungus-stimulating, cellulose-rich amendments to be used in R. solani suppression.

The manuscript is very well written, clear and very well organized. The abstract is concise and clear. The Introduction presents a good background of the research subject. The objectives of the study are very well defined and the hypotheses are carefully well identified. The Materials and Methods section describes clearly the methodologies and the authors provide an excellent and detailed statistical analysis. The Results are clear and the graphics and images are well and clearly presented. The authors make a good discussion of the results and the conclusions are justified and supported by the results.

Specific comments

  • Line 308: ‘sampleswere’ include space between
  • Line 321: include the amount of template DNA in nanograms

Author Response

In the manuscript ‘Impact of cellulose-rich organic soil amendments on growth dynamics and pathogenicity of Rhizoctonia solani’ Clocchiatti and collaborators investigated the potential of fungus-stimulating, cellulose-rich amendments to be used in R. solani suppression.

The manuscript is very well written, clear and very well organized. The abstract is concise and clear. The Introduction presents a good background of the research subject. The objectives of the study are very well defined and the hypotheses are carefully well identified. The Materials and Methods section describes clearly the methodologies and the authors provide an excellent and detailed statistical analysis. The Results are clear and the graphics and images are well and clearly presented. The authors make a good discussion of the results and the conclusions are justified and supported by the results.

Specific comments

  • Line 308: ‘sampleswere’ include space between
  • Line 321: include the amount of template DNA in nanograms

Response: we thank the reviewer for the positive feedback. We made the suggested corrections at line 308 an 321. Please see the attachment.

Reviewer 2 Report

This is the best-written manuscript I had a privilege to read in years - Congratulations!

Author Response

This is the best-written manuscript I had a privilege to read in years - Congratulations!

 Response: we thank the reviewer for the positive feedback. Modifications can be seen in the attachment.

Reviewer 3 Report

Several typos corrections is needed, clarification and additions are also needed. Reduction in the lenght and improving the clearness of the manuscript must be done in order to increase its readibility and understanding it much better. A partly annotated version is enclosed.

Author Response

Several typos corrections is needed, clarification and additions are also needed. Reduction in the lenght and improving the clearness of the manuscript must be done in order to increase its readibility and understanding it much better. A partly annotated version is enclosed.

Response: We thank the reviewer for the detailed comments. The manuscript was corrected and shortened according to the suggestions provided by the reviewer in the annotated version. Please see the changes in the attachment. 

L63: Decomposable should be kept in, as very recalcitrant organic matter (like in former peat soils) do no support growth of fungi.

Results: Italicized only latin terms indicating family. From https://wwwnc.cdc.gov/eid/page/scientific-nomenclature:

‘Italicize family, genus, species, and variety or subspecies. Begin family and genus with a capital letter. Kingdom, phylum, class, order, and suborder begin with a capital letter but are not italicized. If a generic plural for an organism exists (see Dorland’s), it is neither capitalized nor italicized.’

Lines 611-612: since not all readers go through the whole article line by line, we would prefer mentioning briefly this key point both in the introduction and in the discussion

Line 663: rather than moving large portions of the section 4.3 in section 4.2, we moved ll 645-663 at the end of section 4.2, immediately before section 4.3, so that as suggested the further examination of the effect of paper pulp follows immediately in the discussion.

Lines 676-678: the presence of compounds of oak and elder sawdust are discussed in section 4.1

Reviewer 4 Report

I read with great interest the paper by Clocchiatti et al.
The present work focuses on the responses of the phytopathogenic fungus
Rhizoctonia solani in presence of cellulose rich amendments in terms of
biomass and pathogenicity.
T
he paper is well written, pleasant to read, the experiments are
planned properly and the methods are well described. The results are clearly
displayed with the support of very informative charts and tables. The data
are critically discussed in the light of an extensive literature.
This is may be the first time that
I don't find any critical point to
discuss with the authors.
I have only very minor revisions to suggest:
1)line 54: a , is missing after however
2) line 154: immobilization
3)line 156: please define WHC
4) Table 1: please write Rhizoctonia solani in italics
5) line 308: a space is missing between samples and were
6) line 455 and 457: a , is missing before respectively
7)line 569: a space is missing after bacteria

Author Response

I read with great interest the paper by Clocchiatti et al.
The present work focuses on the responses of the phytopathogenic fungus
Rhizoctonia solani in presence of cellulose rich amendments in terms of
biomass and pathogenicity.
The paper is well written, pleasant to read, the experiments are
planned properly and the methods are well described. The results are clearly
displayed with the support of very informative charts and tables. The data
are critically discussed in the light of an extensive literature.
This is may be the first time that I don't find any critical point to
discuss with the authors.
I have only very minor revisions to suggest:
1)line 54: a , is missing after however
2) line 154: immobilization
3)line 156: please define WHC
4) Table 1: please write Rhizoctonia solani in italics
5) line 308: a space is missing between samples and were
6) line 455 and 457: a , is missing before respectively
7)line 569: a space is missing after bacteria

Response: we thank the reviewer for the positive feedback. The suggested revisions  have been incorporated in the new version of the manuscript. Please see the attachment.

Reviewer 5 Report

Abstract: too long for me, presents too many results

L36-37: keywords alphabetically

L68-69: references?

L104-124: the part about hypotheses is a bit too long, I suggest shortening it, making it more precise. L112-124 are really unnecessary, because there should be no description of the experiment here.

L127-137: It is worth separating the two experiences clearly and describing them independently. So as not to switch between the first and the second in one text.  The table can be one, just refer to it twice. As in the results are described these experiments independently.

L165-175: and from what depth was the soil taken?

L204-220: justify the text

L254-257: superscript in unit

Figure 1 and 2: add to the caption that this is a study/photographs by the authors

L314, 328: provide sequences of primers, can be listed in a table of sequences, primer name and references

L322: R. solani in italics

L334: reference to R.

L379-390: this is not a statistical analysis but a bioinformatics or sequencing data analysis, please extract in another subchapter; have the obtained sequences been deposited in some database? If so - provide a reference, if not I suggest you do so. This increases the coverage of the results and adds credibility to the research.

Figure 4, 5, 6: please add with which test the differences were analysed

Figure 7: lack of caption for the x-axis

Figure 8: no caption for the x-axis; besides, maybe it would be better to change the unit range for control and oak to make the data more readable? Were they actually 0?

L598: some confusion with spaces in this sentence

L642-643, 695-696: italics

L1012: unnecessary number 1

Tabe S1, Table S4, Figure S1: specify which test used to analyse the differences

Figure S2 and S3: for each graph came out with the same values of PCoA coefficients 1 and 2? It is useful to sign the graphs from A to F and caption them.

Figure S4 and S5: in my opinion the axes should be signed in each graph, not only in the last one.

Figure S6, S7, S8: in the legend we have "class" and in the description "families" - in the end what do we have on the graph or what should be?

Author Response

Abstract: too long for me, presents too many results

Response: We thank the reviewer for the detailed comments. The result section of the abstract has been changed, by rearranging sentences and removing some details.

All changes can be seen in the attachment - the supplementary materials are included at the end of the document.

L36-37: keywords alphabetically

Response: keywords have been rearranged alphabetically

L68-69: references?

Response: references have been added

L104-124: the part about hypotheses is a bit too long, I suggest shortening it, making it more precise.

Response: the paragraph on hypotheses has been rephrased

L112-124 are really unnecessary, because there should be no description of the experiment here.

Response: lines 112 – 124 have been removed

eL127-137: It is worth separating the two experiences clearly and describing them independently. So as not to switch between the first and the second in one text.  The table can be one, just refer to it twice. As in the results are described these experiments independently.

Response: We thank the reviewer for this suggestion. Each experiment was now introduced independently, with reference to Table 1 repeated when needed.                                                                      

L165-175: and from what depth was the soil taken?

Response: it was indicated in line 170, now it has been rephrased more clearly.

L204-220: justify the text

Response: corrected

L254-257: superscript in unit

Response: corrected

Figure 1 and 2: add to the caption that this is a study/photographs by the authors

Response: added to figure 1. Figure 2 describes the experimental design, which is clearly done by the authors.

L314, 328: provide sequences of primers, can be listed in a table of sequences, primer name and references

Response: the sequences of primers has been added within text.

L322: R. solani in italics

Response: corrected

L334: reference to R.

Response: added

L379-390: this is not a statistical analysis but a bioinformatics or sequencing data analysis, please extract in another subchapter; have the obtained sequences been deposited in some database? If so - provide a reference, if not I suggest you do so. This increases the coverage of the results and adds credibility to the research.

Response: the title of the section was changed into ‘bioinformatic and statistical analysis’. Yes, all the data has been deposited in a database. The references are provided in the Data Availability Statement of the manuscript.

Figure 4, 5, 6: please add with which test the differences were analysed

Response: added

Figure 7: lack of caption for the x-axis

Response: added

Figure 8: no caption for the x-axis; besides, maybe it would be better to change the unit range for control and oak to make the data more readable? Were they actually 0?

Response: the x-axis label ‘day’ has been added. The values for oak were either zero or very close to zero (Table S4), moreover there was no significant effect of oak on the abundance of R. solani in soil(Table S4).

L598: some confusion with spaces in this sentence

Response: adjusted.

L642-643, 695-696: italics

Response: adjusted.

L1012: unnecessary number 1

Response: removed

Tabe S1, Table S4, Figure S1: specify which test used to analyse the differences

Response: added

Figure S2 and S3: for each graph came out with the same values of PCoA coefficients 1 and 2? It is useful to sign the graphs from A to F and caption them.

Response: we apologize for the confusion. Figure S2 represents one PCoA analysis (caption: “The results of the same PCoA ordination analysis (based on a Bray-Curtis distances) are plotted separately for soil sampled at day 7, 21, 35, 49 and for root samples.”), as well as figure S3. The samples are represented in separate plots for clarity (i.e. avoiding having many samples in a plot or a complex color coding).

Figure S4 and S5: in my opinion the axes should be signed in each graph, not only in the last one.

Response: the axis labels have been added

Figure S6, S7, S8: in the legend we have "class" and in the description "families" - in the end what do we have on the graph or what should be?

Response: in the graph individual dots represent families of bacteria / fungi, which are distributed along the y-axis based on the log2 fold change (under/overrepresented). For clarity, families are color coded and arranged along the x-axis by class. The caption has been slightly rephrased.

Round 2

Reviewer 3 Report

No major comments